

# Automated language essay scoring systems: a literature review

Mohamed Abdellatif Hussein[1], Hesham Hassan[2] and Mohammad Nassef[2]

[1] Information and Operations, National Center for Examination and Educational Evaluation, Cairo, Egypt
[2] Faculty of Computers and Information, Computer Science Department, Cairo University, Cairo, Egypt

## ABSTRACT

**Background**. Writing composition is a significant factor for measuring test-takers' ability in any language exam. However, the assessment (scoring) of these writing compositions or essays is a very challenging process in terms of reliability and time. The need for objective and quick scores has raised the need for a computer system that can automatically grade essay questions targeting specific prompts. Automated Essay Scoring (AES) systems are used to overcome the challenges of scoring writing tasks by using Natural Language Processing (NLP) and machine learning techniques. The purpose of this paper is to review the literature for the AES systems used for grading the essay questions.

**Methodology**. We have reviewed the existing literature using Google Scholar, EBSCO and ERIC to search for the terms "AES", "Automated Essay Scoring", "Automated Essay Grading", or "Automatic Essay" for essays written in English language. Two categories have been identified: handcrafted features and automatically featured AES systems. The systems of the former category are closely bonded to the quality of the designed features. On the other hand, the systems of the latter category are based on the automatic learning of the features and relations between an essay and its score without any handcrafted features. We reviewed the systems of the two categories in terms of system primary focus, technique(s) used in the system, the need for training data, instructional application (feedback system), and the correlation between e-scores and human scores. The paper includes three main sections. First, we present a structured literature review of the available Handcrafted Features AES systems. Second, we present a structured literature review of the available Automatic Featuring AES systems. Finally, we draw a set of discussions and conclusions.

**Results**. AES models have been found to utilize a broad range of manually-tuned shallow and deep linguistic features. AES systems have many strengths in reducing labor-intensive marking activities, ensuring a consistent application of scoring criteria, and ensuring the objectivity of scoring. Although many techniques have been implemented to improve the AES systems, three primary challenges have been identified. The challenges are lacking of the sense of the rater as a person, the potential that the systems can be deceived into giving a lower or higher score to an essay than it deserves, and the limited ability to assess the creativity of the ideas and propositions and evaluate their practicality. Many techniques have only been used to address the first two challenges.

Corresponding author
Mohamed Abdellatif Hussein,
teeefa@nceee.edu.eg,
teeefa@gmail.com

# INTRODUCTION

Test items (questions) are usually classified into two types: selected-response (SR), and constructed-response (CR). The SR items, such as true/false, matching or multiple-choice, are much easier than the CR items in terms of objective scoring (*Isaacs et al., 2013*). SR questions are commonly used for gathering information about knowledge, facts, higher-order thinking, and problem-solving skills. However, considerable skill is required to develop test items that measure analysis, evaluation, and other higher cognitive skills (*Stecher et al., 1997*).

CR items, sometimes called open-ended, include two sub-types: restricted-response and extended-response items (*Nitko & Brookhart, 2007*). Extended-response items, such as essays, problem-based examinations, and scenarios, are like restricted-response items, except that they extend the demands made on test-takers to include more complex situations, more difficult reasoning, and higher levels of understanding which are based on real-life situations requiring test-takers to apply their knowledge and skills to new settings or situations (*Isaacs et al., 2013*).

In language tests, test-takers are usually required to write an essay about a given topic. Human-raters score these essays based on specific scoring rubrics or schemes. It occurs that the score of an essay scored by different human-raters vary substantially because human scoring is subjective (*Peng, Ke & Xu, 2012*). As the process of human scoring takes much time, effort, and are not always as objective as required, there is a need for an automated essay scoring system that reduces cost, time and determines an accurate and reliable score.

Automated Essay Scoring (AES) systems usually utilize Natural Language Processing and machine learning techniques to automatically rate essays written for a target prompt (*Dikli, 2006*). Many AES systems have been developed over the past decades. They focus on automatically analyzing the quality of the composition and assigning a score to the text. Typically, AES models exploit a wide range of manually-tuned shallow and deep linguistic features (*Farag, Yannakoudakis & Briscoe, 2018*). Recent advances in the deep learning approach have shown that applying neural network approaches to AES systems has accomplished state-of-the-art results (*Page, 2003*; *Valenti, Neri & Cucchiarelli, 2017*) with the additional benefit of using features that are automatically learnt from the data.

## Survey methodology

The purpose of this paper is to review the AES systems literature pertaining to scoring extended-response items in language writing exams. Using Google Scholar, EBSCO and ERIC, we searched the terms ''AES'', ''Automated Essay Scoring'', ''Automated Essay Grading'', or ''Automatic Essay'' for essays written in English language. AES systems which score objective or restricted-response items are excluded from the current research.

The most common models found for AES systems are based on Natural Language Processing (NLP), Bayesian text classification, Latent Semantic Analysis (LSA), or Neural Networks. We have categorized the reviewed AES systems into two main categories. The former is based on handcrafted discrete features bounded to specific domains. The latter

is based on automatic feature extraction. For instance, Artificial Neural Network (ANN)-based approaches are capable of automatically inducing dense syntactic and semantic features from a text.

The literature of the two categories has been structurally reviewed and evaluated based on certain factors including: system primary focus, technique(s) used in the system, the need for training data, instructional application (feedback system), and the correlation between e-scores and human scores.

## Handcrafted features AES systems
### Project Essay Grader<sup>TM</sup> (PEG)

Ellis Page developed the PEG in 1966. PEG is considered the earliest AES system that has been built in this field. It utilizes correlation coefficients to predict the intrinsic quality of the text. It uses the terms "trins" and "proxes" to assign a score. Whereas "trins" refers to intrinsic variables like diction, fluency, punctuation, and grammar, "proxes" refers to correlations between intrinsic variables such as average length of words in a text, and/or text length. (*Dikli, 2006*; *Valenti, Neri & Cucchiarelli, 2017*).

The PEG uses a simple scoring methodology that consists of two stages. The former is the training stage and the latter is the scoring stage. PEG should be trained on a sample of essays from 100 to 400 essays, the output of the training stage is a set of coefficients ($\beta$ weights) for the proxy variables from the regression equation. In the scoring stage, proxes are identified for each essay, and are inserted into the prediction equation. To end, a score is determined by estimating coefficients ($\beta$ weights) from the training stage (*Dikli, 2006*).

Some issues have been marked as a criticism for the PEG such as disregarding the semantic side of essays, focusing on surface structures, and not working effectively in case of receiving student responses directly (which might ignore writing errors). PEG has a modified version released in 1990, which focuses on grammar checking with a correlation between human assessors and the system ($r = 0.87$) (*Dikli, 2006*; *Page, 1994*; *Refaat, Ewees & Eisa, 2012*).

Measurement Inc. acquired the rights of PEG in 2002 and continued to develop it. The modified PEG analyzes the training essays and calculates more than 500 features that reflect intrinsic characteristics of writing, such as fluency, diction, grammar, and construction. Once the features have been calculated, the PEG uses them to build statistical and linguistic models for the accurate prediction of essay scores (*Home—Measurement Incorporated, 2019*).

### Intelligent Essay Assessor<sup>TM</sup> (IEA)

IEA was developed by *Landauer (2003)*. IEA uses a statistical combination of several measures to produce an overall score. It relies on using Latent Semantic Analysis (LSA); a machine-learning model of human understanding of the text that depends on the training and calibration methods of the model and the ways it is used tutorially (*Dikli, 2006*; *Foltz, Gilliam & Kendall, 2003*; *Refaat, Ewees & Eisa, 2012*).

IEA can handle students' innovative answers by using a mix of scored essays and the domain content text in the training stage. It also spots plagiarism and provides feedback (*Dikli, 2006*; *Landauer, 2003*). It uses a procedure for assigning scores in a process that

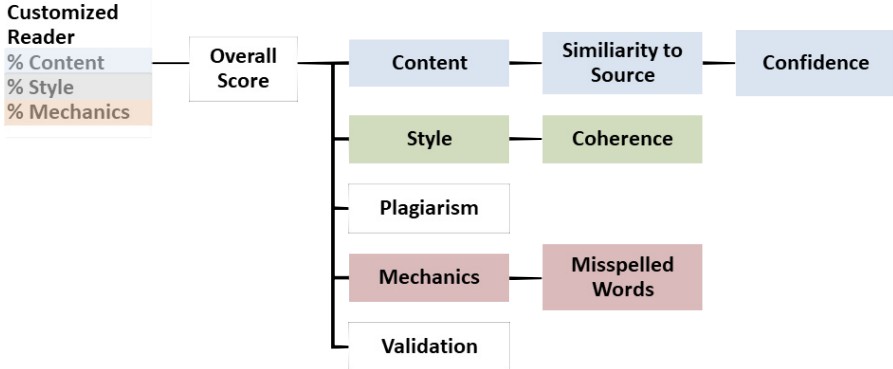

**Figure 1  The IEA architecture.**

begins with comparing essays to each other in a set. LSA examines the extremely similar essays. Irrespective of the replacement of paraphrasing, synonym, or reorganization of sentences, the two essays will be similar LSA. Plagiarism is an essential feature to overcome academic dishonesty, which is difficult to be detected by human-raters, especially in the case of grading a large number of essays (*Dikli, 2006*; *Landauer, 2003*). (Fig. 1) represents IEA architecture (*Landauer, 2003*). IEA requires smaller numbers of pre-scored essays for training. On the contrary of other AES systems, IEA requires only 100 pre-scored training essays per each prompt vs. 300–500 on other systems (*Dikli, 2006*).

*Landauer (2003)* used IEA to score more than 800 students' answers in middle school. The results showed a 0.90 correlation value between IEA and the human-raters. He explained the high correlation value due to several reasons including that human-raters could not compare each essay to each other for the 800 students while IEA can do so (*Dikli, 2006*; *Landauer, 2003*).

### E-rater®

Educational Testing Services (ETS) developed E-rater in 1998 to estimate the quality of essays in various assessments. It relies on using a combination of statistical and NLP techniques to extract linguistic features (such as grammar, usage, mechanics, development) from text to start processing, then compares scores with human graded essays (*Attali & Burstein, 2014*; *Dikli, 2006*; *Ramineni & Williamson, 2018*).

The E-rater system is upgraded annually. The current version uses 11 features divided into two areas: writing quality (grammar, usage, mechanics, style, organization, development, word choice, average word length, proper prepositions, and collocation usage), and content or use of prompt-specific vocabulary (*Ramineni & Williamson, 2018*).

The E-rater scoring model consists of two stages: the model of the training stage, and the model of the evaluation stage. Human scores are used for training and evaluating the E-rater scoring models. The quality of the E-rater models and its effective functioning in an operational environment depend on the nature and quality of the training and evaluation

data (*Williamson, Xi & Breyer, 2012*). The correlation between human assessors and the system ranged from 0.87 to 0.94 (*Refaat, Ewees & Eisa, 2012*).

### Criterion^SM

Criterion is a web-based scoring and feedback system based on ETS text analysis tools: E-rater® and Critique. As a text analysis tool, Critique integrates a collection of modules that detect faults in usage, grammar, and mechanics, and recognizes discourse and undesirable style elements in writing. It provides immediate holistic scores as well (*Crozier & Kennedy, 1994*; *Dikli, 2006*).

Criterion similarly gives personalized diagnostic feedback reports based on the types of assessment instructors give when they comment on students' writings. This component of the Criterion is called an advisory component. It is added to the score, but it does not control it[18]. The types of feedback the advisory component may provide are like the following:

- The text is too brief (a student may write more).
- The essay text does not look like other essays on the topic (the essay is off-topic).
- The essay text is overly repetitive (student may use more synonyms) (*Crozier & Kennedy, 1994*).

### IntelliMetric^TM

Vantage Learning developed the IntelliMetric systems in 1998. It is considered the first AES system which relies on Artificial Intelligence (AI) to simulate the manual scoring process carried out by human-raters under the traditions of cognitive processing, computational linguistics, and classification (*Dikli, 2006*; *Refaat, Ewees & Eisa, 2012*).

IntelliMetric relies on using a combination of Artificial Intelligence (AI), Natural Language Processing (NLP) techniques, and statistical techniques. It uses CogniSearch and Quantum Reasoning technologies that were designed to enable IntelliMetric to understand the natural language to support essay scoring (*Dikli, 2006*).

IntelliMetric uses three steps to score essays as follows:

a) First, the training step that provides the system with known scores essays.
b) Second, the validation step examines the scoring model against a smaller set of known scores essays.
c) Finally, application to new essays with unknown scores. (*Learning, 2000*; *Learning, 2003*; *Shermis & Barrera, 2002*)

IntelliMetric identifies text related characteristics as larger categories called Latent Semantic Dimensions (LSD). (Figure 2) represents the IntelliMetric features model.

IntelliMetric scores essays in several languages including English, French, German, Arabic, Hebrew, Portuguese, Spanish, Dutch, Italian, and Japanese (*Elliot, 2003*). According to Rudner, Garcia, and Welch (*Rudner, Garcia & Welch, 2006*), the average of the correlations between IntelliMetric and human-raters was 0.83 (*Refaat, Ewees & Eisa, 2012*).

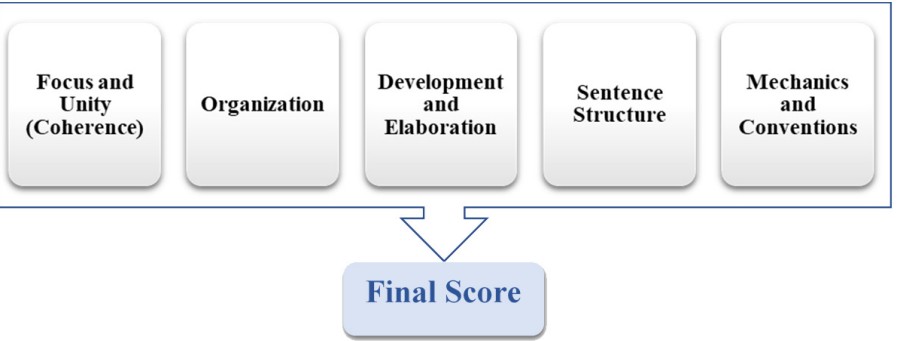

**Figure 2  The IntelliMetric features model.**

### MY Access!

MY Access is a web-based writing assessment system based on the IntelliMetric AES system. The primary aim of this system is to provide immediate scoring and diagnostic feedback for the students' writings in order to motivate them to improve their writing proficiency on the topic (*Dikli, 2006*).

MY Access system contains more than 200 prompts that assist in an immediate analysis of the essay. It can provide personalized Spanish and Chinese feedback on several genres of writing such as narrative, persuasive, and informative essays. Moreover, it provides multilevel feedback—developing, proficient, and advanced—as well (*Dikli, 2006*; *Learning, 2003*).

### Bayesian Essay Test Scoring System<sup>TM</sup> (BETSY)

BETSY classifies the text based on trained material. It has been developed in 2002 by Lawrence Rudner at the College Park of the University of Maryland with funds from the US Department of Education (*Valenti, Neri & Cucchiarelli, 2017*). It has been designed to automate essay scoring, but can be applied to any text classification task (*Taylor, 2005*).

BETSY needs to be trained on a huge number (1,000 texts) of human classified essays to learn how to classify new essays. The goal of the system is to determine the most likely classification of an essay to a set of groups (Pass-Fail) and (Advanced - Proficient - Basic - Below Basic) (*Dikli, 2006*; *Valenti, Neri & Cucchiarelli, 2017*). It learns how to classify a new document through the following steps:

The first-word training step is concerned with the training of words, evaluating database statistics, eliminating infrequent words, and determining stop words.

The second-word pairs training step is concerned with evaluating database statistics, eliminating infrequent word-pairs, maybe scoring the training set, and trimming misclassified training sets.

Finally, BETSY can be applied to a set of experimental texts to identify the classification precision for several new texts or a single text. (*Dikli, 2006*)

BETSY has achieved accuracy of over 80%, when trained with 462 essays, and tested with 80 essays (*Rudner & Liang, 2002*).

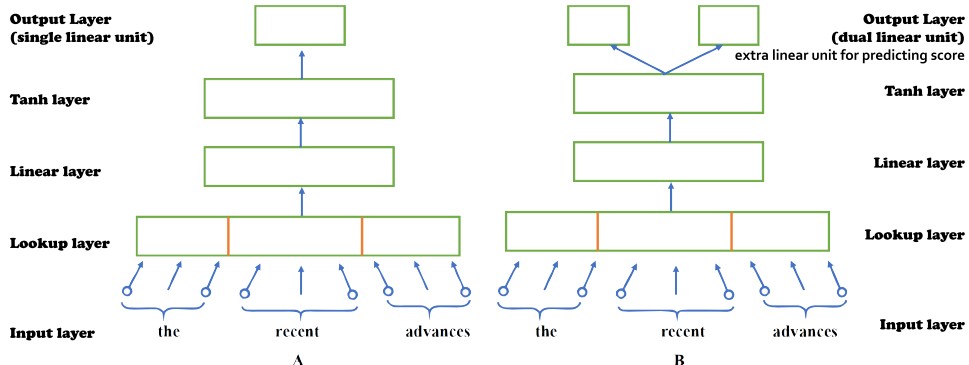

**Figure 3** **The architectures of two models.** (A) Original C&W model. (B) Augmented C&W model.

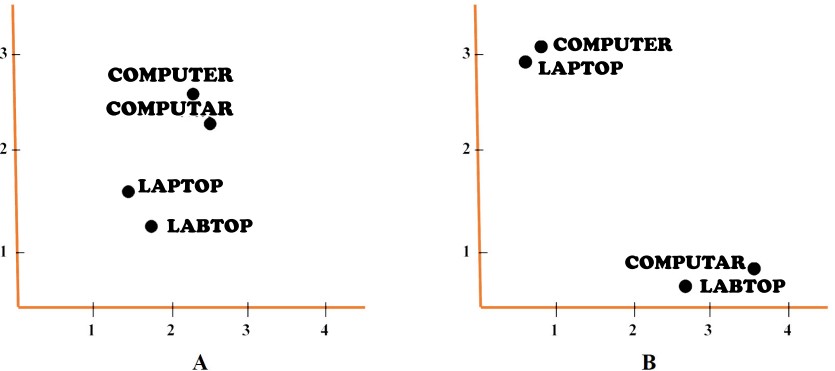

**Figure 4** **The example of embeddings.** (A) Standard neural embeddings. (B) *SSWE* word embeddings.

## Automatic featuring AES systems
### Automatic text scoring using neural networks

Alikaniotis, Yannakoudakis, and Rei introduced in 2016 a deep neural network model capable of learning features automatically to score essays. This model has introduced a novel method to identify the more discriminative regions of the text using: (1) a Score-Specific Word Embedding (SSWE) to represent words and (2) a two-layer Bidirectional Long-Short-Term Memory (LSTM) network to learn essay representations. (*Alikaniotis, Yannakoudakis & Rei, 2016*; *Taghipour & Ng, 2016*).

Alikaniotis and his colleagues have extended the *C&W Embeddings* model into the *Augmented C&W* model to capture, not only the local linguistic environment of each word, but also how each word subsidizes to the overall score of an essay. In order to capture *SSWEs*. A further linear unit has been added in the output layer of the previous model which performs linear regression, predicting the essay score (*Alikaniotis, Yannakoudakis & Rei, 2016*). Figure 3 shows the architectures of two models, (A) Original C&W model and (B) Augmented C&W model. Figure 4 shows the example of (A) standard neural embeddings to (B) *SSWE* word embeddings.

SSWEs obtained by their model used to derive continuous representations for each essay. Each essay is identified as a sequence of tokens. The uni- and bi-directional LSTMs have been efficiently used for embedding long sequences (*Alikaniotis, Yannakoudakis & Rei, 2016*).

They used the Kaggle's ASAP (https://www.kaggle.com/c/asap-aes/data) contest dataset. It consists of 12.976 essays, with average length 150-to-550 words per essay, each double marked (Cohen's = 0.86). The essays presented eight different prompts, each with distinct marking criteria and score range.

Results showed that SSWE and the LSTM approach, without any prior knowledge of the language grammar or the text domain, was able to mark the essays in a very human-like way, beating other state-of-the-art systems. Furthermore, while tuning the models' hyperparameters on a separate validation set (*Alikaniotis, Yannakoudakis & Rei, 2016*), they did not perform any further preprocessing of the text other than simple tokenization. Also, it outperforms the traditional SVM model by combining SSWE and LSTM. On the contrary, LSTM alone did not give significant more accuracies compared to SVM.

According to Alikaniotis, Yannakoudakis, and Rei (*Alikaniotis, Yannakoudakis & Rei, 2016*), the combination of SSWE with the two-layer bi-directional LSTM had the highest correlation value on the test set averaged 0.91 (Spearman) and 0.96 (Pearson).

### A neural network approach to automated essay scoring

Taghipour and H. T. Ng developed in 2016 a Recurrent Neural Networks (RNNs) approach which automatically learns the relation between an essay and its grade. Since the system is based on RNNs, it can use non-linear neural layers to identify complex patterns in the data and learn them, and encode all the information required for essay evaluation and scoring (*Taghipour & Ng, 2016*).

The designed model architecture can be presented in five layers as follow:

a) The Lookup Table Layer; which builds $d_{LT}$ dimensional space containing each word projection.

b) The Convolution Layer; which extracts feature vectors from n-grams. It can possibly capture local contextual dependencies in writing and therefore enhance the performance of the system.

c) The Recurrent Layer; which processes the input to generate a representation for the given essay.

d) The Mean over Time; which aggregates the variable number of inputs into a fixed length vector.

e) The Linear Layer with Sigmoid Activation; which maps the generated output vector from the mean-over-time layer to a scalar value (*Taghipour & Ng, 2016*).

Taghipour and his colleagues have used the Kaggle's ASAP contest dataset. They distributed the data set into 60% training set, 20% a development set, and 20% a testing set. They used Quadratic Weighted Kappa (QWK) as an evaluation metric. For evaluating the performance of the system, they compared it to an available open source AES system called the 'Enhanced AI Scoring Engine' (EASE) (https://github.com/edx/ease). To identify the best model, they performed several experiments like Convolutional vs. Recurrent

Neural Network, basic RNN vs. Gated Recurrent Units (GRU) vs. LSTM, unidirectional vs. Bidirectional LSTM, and using with vs. without mean-over-time layer (*Taghipour & Ng, 2016*).

The results showed multiple observations according to (*Taghipour & Ng, 2016*), summarized as follows:

a) RNN failed to get accurate results as LSTM or GRU and the other models outperformed it. This was possibly due to the relatively long sequences of words in writing.

b) The neural network performance was significantly affected with the absence of the mean over-time layer. As a result, it did not learn the task in an exceedingly proper manner.

c) The best model was the combination of ten instances of LSTM models with ten instances of CNN models. The new model outperformed the baseline EASE system by 5.6% and with averaged QWK value 0.76.

### Automatic features for essay scoring—an empirical study

Dong and Zhang provided in 2016 an empirical study to examine a neural network method to learn syntactic and semantic characteristics automatically for AES, without the need for external pre-processing. They built a hierarchical Convolutional Neural Network (CNN) structure with two levels in order to model sentences separately (*Dasgupta et al., 2018*; *Dong & Zhang, 2016*).

Dong and his colleague built a model with two parts, summarized as follows:

a) Word Representations: A word embedding is used but does not rely on POS-tagging or other pre-processing.

b) CNN Model: They took essay scoring as a regression task and employed a two-layer CNN model, in which one Convolutional layer is used to extract sentences representations, and the other is stacked on sentence vectors to learn essays representations.

The dataset that they employed in experiments is the Kaggle's ASAP contest dataset. The settings of data preparation followed the one that Phandi, Chai, and Ng used (*Phandi, Chai & Ng, 2015*). For domain adaptation (cross-domain) experiments, they followed Phandi, Chai, and Ng (*Phandi, Chai & Ng, 2015*), by picking four pairs of essay prompts, namely, $1 \rightarrow 2$, $3 \rightarrow 4$, $5 \rightarrow 6$ and $7 \rightarrow 8$, where $1 \rightarrow 2$ denotes prompt one as source domain and prompt 2 as target domain. They used quadratic weighted Kappa (QWK) as the evaluation metric.

In order to evaluate the performance of the system, they compared it to EASE system (an open source AES available for public) with its both models Bayesian Linear Ridge Regression (BLRR) and Support Vector Regression (SVR).

The Empirical results showed that the two-layer Convolutional Neural Network (CNN) outperformed other baselines (e.g., Bayesian Linear Ridge Regression) on both in-domain and domain adaptation experiments on the Kaggle's ASAP contest dataset. So, the neural features learned by CNN were very effective in essay marking, handling more high-level and abstract information compared to manual feature templates. In domain average, QWK value was 0.73 vs. 0.75 for human rater (*Dong & Zhang, 2016*).

### Augmenting textual qualitative features in deep convolution recurrent neural network for automatic essay scoring

In 2018, *Dasgupta et al. (2018)* proposed a Qualitatively enhanced Deep Convolution Recurrent Neural Network architecture to score essays automatically. The model considers both word- and sentence-level representations. Using a Hierarchical CNN connected with a Bidirectional LSTM model they were able to consider linguistic, psychological and cognitive feature embeddings within a text (*Dasgupta et al., 2018*).

The designed model architecture for the linguistically informed Convolution RNN can be presented in five layers as follow:

a) Generating Embeddings Layer: The primary function is constructing previously trained sentence vectors. Sentence vectors extracted from every input essay are appended with the formed vector from the linguistic features determined for that sentence.

b) Convolution Layer: For a given sequence of vectors with K windows, this layer function is to apply linear transformation for all these K windows. This layer is fed by each of the generated word embeddings from the previous layer.

c) Long Short-Term Memory Layer: The main function of this layer is to examine the future and past sequence context by connecting Bidirectional LSTMs (Bi-LSTM) networks.

d) Activation layer: The main function of this layer is to obtain the intermediate hidden layers from the Bi-LSTM layer $h_1$, $h_2$,..., $h_T$, and in order to calculate the weights of sentence contribution to the final essay's score (quality of essay). They used an attention pooling layer over sentence representations.

e) The Sigmoid Activation Function Layer: The main function of this layer is to perform a linear transformation of the input vector that converts it to a scalar value (continuous) (*Dasgupta et al., 2018*).

Figure 5 represents the proposed linguistically informed Convolution Recurrent Neural Network architecture.

Dasgupta and his colleagues employed in their experiments the Kaggle's ASAP contest dataset. They have done 7 folds using cross validation technique to assess their models. Every fold is distributed as follows; training set which represents 80% of the data, development set represented by 10%, and the rest 10% as the test set. They used quadratic weighted Kappa (QWK) as the evaluation metric.

The results showed that, in terms of all these parameters, the Qualitatively Enhanced Deep Convolution LSTM (Qe-C-LSTM) system performed better than the existing, LSTM, Bi-LSTM and EASE models. It achieved a Pearson's and Spearman's correlation of 0.94 and 0.97 respectively as compared to that of 0.91 and 0.96 in *Alikaniotis, Yannakoudakis & Rei (2016)*. They also accomplished an RMSE score of 2.09. They computed a pairwise Cohen's k value of 0.97 as well (*Dasgupta et al., 2018*).

## SUMMARY AND DISCUSSION

Over the past four decades, there have been several studies that examined the approaches of applying computer technologies on scoring essay questions. Recently, computer

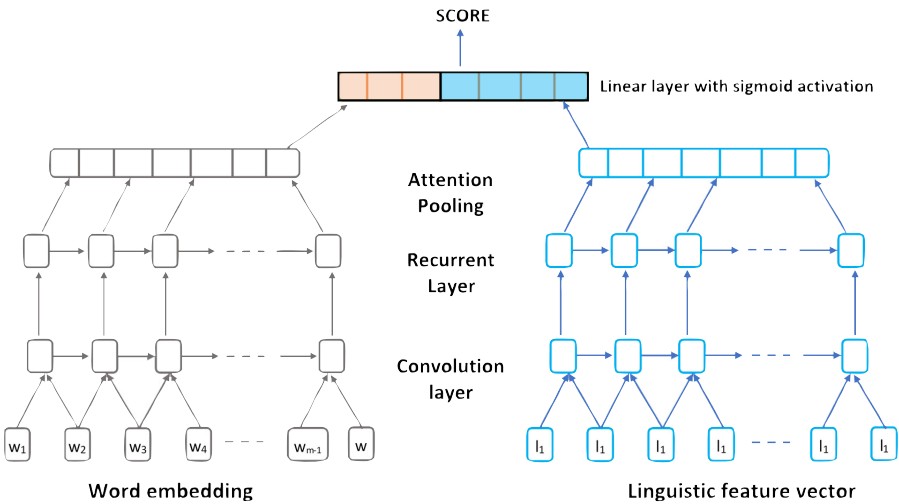

**Figure 5** **The proposed linguistically informed Convolution Recurrent Neural Network architecture.**

technologies have been able to assess the quality of writing using AES technology. Many attempts have taken place in developing AES systems in the past years (*Dikli, 2006*).

The AES systems do not assess the intrinsic qualities of an essay directly as human-raters do, but they utilize the correlation coefficients of the intrinsic qualities to predict the score to be assigned to an essay. The performance of these systems is evaluated based on the comparison of the scores assigned to a set of essays scored by expert humans.

The AES systems have many strengths mainly in reducing labor-intensive marking activities, overcoming time, cost, and improving the reliability of writing tasks. Besides, they ensure a consistent application of marking criteria, therefore facilitating equity in scoring. However, there is a substantial manual effort involved in reaching these results on different domains, genres, prompts, and so forth. Moreover, the linguistic features intended to capture the aspects of writing to be assessed are hand-selected and tuned for specific domains. In order to perform well on different data, separate models with distinct feature sets are typically tuned (*Burstein, 2003*; *Dikli, 2006*; *Hamp-Lyons, 2001*; *Rudner & Gagne, 2001*; *Rudner & Liang, 2002*). Despite its weaknesses, the AES systems continue to attract the attention of public schools, universities, testing agencies, researchers and educators (*Dikli, 2006*).

The AES systems described in this paper under the first category are based on handcrafted features and, usually, rely on regression methods. They employ several methods to obtain the scores. While E-rater and IntelliMetric use NLP techniques, the IEA system utilizes LSA. Moreover, PEG utilizes proxy measures (proxes), and BETSY[TM] uses Bayesian procedures to evaluate the quality of a text.

While E-rater, IntelliMetric, and BETSY evaluate style and semantic content of essays, PEG only evaluates style and ignores the semantic aspect of essays. Furthermore, IEA is exclusively concerned with semantic content. Unlike PEG, E-rater, IntelliMetric, and IEA

need smaller numbers of pre-scored essays for training in contrast with BETSY which needs a huge number of training pre-scored essays.

The systems in the first category have high correlations with human-raters. While PEG, E-rater, IEA, and BETSY evaluate only English language essay responses, IntelliMetric evaluates essay responses in multiple languages.

Contrary to PEG, IEA, and BETSY, E-rater, and IntelliMetric have instructional or immediate feedback applications (i.e., Criterion and MY Access!). Instructional-based AES systems have worked hard to provide formative assessments by allowing students to save their writing drafts on the system. Thus, students can review their writings as of the formative feedback received from either the system or the teacher. The recent version of MY Access! (6.0) provides online portfolios and peer review.

The drawbacks of this category may include the following: (a) feature engineering can be time-consuming, since features need to be carefully handcrafted and selected to fit the appropriate model, and (b) such systems are sparse and instantiated by discrete pattern-matching.

AES systems described in this paper under the second category are usually based on neural networks. Neural Networking approaches, especially Deep Learning techniques, have been shown to be capable of inducing dense syntactic and semantic features automatically, applying them to text analysis and classification problems including AES systems (*Alikaniotis, Yannakoudakis & Rei, 2016*; *Dong & Zhang, 2016*; *Taghipour & Ng, 2016*), and giving better results with regards to the statistical models used in the handcrafted features (*Dong & Zhang, 2016*).

Recent advances in Deep Learning have shown that neural approaches to AES achieve state-of-the-art results (*Alikaniotis, Yannakoudakis & Rei, 2016*; *Taghipour & Ng, 2016*) with the additional advantage of utilizing features that are automatically learned from the data. In order to facilitate interpretability of neural models, a number of visualization techniques have been proposed to identify textual (superficial) features that contribute to model performance [7].

While Alikaniotis and his colleagues (*2016*) employed a two-layer Bidirectional LSTM combined with the SSWE for essay scoring tasks, *Taghipour & Ng (2016)* adopted the LSTM model and combined it with CNN. *Dong & Zhang (2016)* developed a two-layer CNN, and Dasgupta and his colleagues (*2018*) proposed a Qualitatively Enhanced Deep Convolution LSTM. Unlike Alikaniotis and his colleagues (*2016*), *Taghipour & Ng (2016)*, *Dong & Zhang (2016)*, Dasgupta and his colleagues (*2018*) were interested in word-level and sentence-level representations as well as linguistic, cognitive and psychological feature embeddings. All linguistic and qualitative features were figured off-line and then entered in the Deep Learning architecture.

Although Deep Learning-based approaches have achieved better performance than the previous approaches, the performance may not be better using the complex linguistic and cognitive characteristics, which are very important in modeling such essays. See Table 1 for the comparison of AES systems.

In general, there are three primary challenges to AES systems. First, they are not able to assess essays as human-raters do because they do what they have been programmed to do

**Table 1  The comparison of AES systems.**

| AES/Parameter | Vendor | Release date | Primary focus | Technique(s) used | Training data | Feedback Application | Correlation with human raters' scores |
|---|---|---|---|---|---|---|---|
| PEG™ | Ellis Page | 1966 | Style | Statistical | Yes (100 – 400) | No | 0.87 |
| IEA™ | Landauer, Foltz, & Laham | 1997 | Content | LSA (KAT engine by PEARSON) | Yes (∼100) | Yes | 0.90 |
| E-rater® | ETS development team | 1998 | Style & Content | NLP | Yes (∼400) | Yes (Criterion) | ∼0.91 |
| IntelliMetric™ | Vantage Learning | 1998 | Style & Content | NLP | Yes (∼300) | Yes (MY Access!) | ∼0.83 |
| BETSY™ | Rudner | 1998 | Style & Content | Bayesian text classification | Yes (1000) | No | ∼0.80 |
| *Alikaniotis, Yannakoudakis & Rei (2016)* | Alikaniotis, Yannakoudakis, and Rei | 2016 | Style & Content | SSWE + Two-layer Bi-LSTM | Yes (∼8000) | No | ∼0.91 (Spearman) ∼0.96 (Pearson) |
| *Taghipour & Ng (2016)* | Taghipour and Ng | 2016 | Style & Content | Adopted LSTM | Yes (∼7786) | NO | QWK for LSTM ∼0.761 |
| *Dong & Zhang (2016)* | Dong and Zhang | 2016 | Syntactic and semantic features | Word embedding and a two-layer Convolution Neural Network | Yes (∼1500 to ∼1800) | NO | average kappa ∼0.734 versus 0.754 for human |
| *Dasgupta et al. (2018)* | Dasgupta, T., Naskar, A., Dey, L., & Saha, R. | 2018 | Style, Content, linguistic and psychological | Deep Convolution Recurrent Neural Network | Yes (∼8000 to 10000) | NO | Pearson's and Spearman's correlation of 0.94 and 0.97 respectively |

**Notes.**
Scorers.

(*Page, 2003*). They eliminate the human element in writing assessment and lack the sense of the rater as a person (*Hamp-Lyons, 2001*). This shortcoming was somehow overcome by obtaining high correlations between the computer and human-raters (*Page, 2003*) although this is still a challenge.

The second challenge is whether the computer can be fooled by students or not (*Dikli, 2006*). It is likely to "trick" the system by writing a longer essay to obtain higher score for example (*Kukich, 2000*). Studies, such as the GRE study in 2001, examined whether a computer could be deceived and assign a lower or higher score to an essay than it should deserve or not. The results revealed that it might reward a poor essay (*Dikli, 2006*). The developers of AES systems have been utilizing algorithms to detect students who try to cheat.

Although automatic learning AES systems based on Neural Networks algorithms, the handcrafted AES systems transcend automatic learning systems in one important feature. Handcrafted systems are highly related to the scoring rubrics that have been designed as a criterion for assessing a specific essay and human-raters use these rubrics to score essays

a well. The objectivity of human-raters is measured by their commitment to the scoring rubrics. On the contrary, automatic learning systems extract the scoring criteria using machine learning and neural networks, which may include some factors that are not part of the scoring rubric, and, hence, is reminiscent of raters' subjectivity (i.e., mode, nature of a rater's character, etc.) Considering this point, handcrafted AES systems may be considered as more objective and fairer to students from the viewpoint of educational assessment.

The third challenge is measuring the creativity of human writing. Accessing the creativity of ideas and propositions and evaluating their practicality are still a pending challenge to both categories of AES systems which still needs further research.

### Funding
The authors received no funding for this work.

### Competing Interests
The authors declare there are no competing interests.

### Author Contributions
- Mohamed Abdellatif Hussein conceived and designed the experiments, performed the experiments, analyzed the data, contributed reagents/materials/analysis tools, prepared figures and/or tables.
- Hesham Hassan and Mohammad Nassef authored or reviewed drafts of the paper, approved the final draft.

### Data Availability
As this is a literature, review, there was no raw data.

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
