# Peer review of "Automated language essay scoring systems: a literature review"

_PeerJ Computer Science, doi:10.7717/peerj-cs.208_

## Round 0.1 · original submission · Minor Revisions

Please take special attention to language issues.

Reviewer 1 ·

Basic reporting

The manuscript brings an excellent review on automated scoring systems, providing the most relevant items in the literature put into context. There is also a fair amount of discussion about the strengths and limitations of existing systems. Overall, the manuscript is likely to be useful for anyone investigating intelligent systems associated with text analytics.

Experimental design

The study was well designed with a systematic search of automated scoring systems, which were then divided into broad categories to facilitate readability. The review is also organized in a logical manner.

Validity of the findings

The findings were reported in a straightforward manner, making it easy for the readers to grasp the most important developments in the field.

Additional comments

This is an excellent review, and my only recommendation is a careful proofreading of the manuscript. There are a few typos that need to be corrected, in addition to inadequate usage of some terms (e.g. "In the other hand", rather than "On the other hand").

Reviewer 2 ·

Basic reporting

The topic of the article is important, and the evaluation of the different approaches/systems is required in the presence of huge use of AES systems especially in English Language Assessment and standardized exam all over the world.

The article has a good introduction and background to demonstrate the importance of the work and how it fits into the field of knowledge. Also, the structure of the article is logical and in an acceptable format.

In general, the language of the article is clear and the ideas are well organized. Yet, there are few English mistakes (See below). I suggest reviewing the article by a native speaker. The tables and figures are clear and useful as well.


In line 185, “the correlations average” to “the average of the correlations”
In line 235, “makred” to “marked”
In line 249, “learn” to “learns”
In line 313, “consider” to “considers”
In line 332, “convert” to “converts”
Line 336, ambiguous sentence. Please rewrite
In line 338, “represent” to “represents”
In line 352, “took” to “taken”
In line 380, “On contrary of” to “On the contrary to”
In line 391, delete “and” from “…… automatically, and apply them ……”
In line 435, delete “to AES systems” in “The third challenge to AES systems is measuring ….“

Experimental design

The article content is within the aims and scope of the journal. The method is well described and is sufficient for the purpose and nature of the article. It can be reproduced. The survey method is unbiased. The review is organized logically into coherent sections.

It seems that the focus of the article is AES systems that evaluate English language writings only. I think it will be an enrichment if it includes also systems for other languages such as Arabic; (For example Al-Jouie, M. F., & Azmi, A. M. (2017). Automated Evaluation of School Children Essays in Arabic. Procedia Computer Science, 117, 19–22. https://doi.org/10.1016/j.procs.2017.10.089). Otherwise, it should be stated clearly as a limitation of the article. It is up to the authors.

I also propose to include another system from China: Yubing, Q. (2016). Pigai Smart Essay Scoring System and Its Implications for Teaching English Writing. Journal of Applied Science and Engineering Innovation, 3(6), 217–219. Retrieved from http://www.jyb.cn/high/gdjyxw/

Validity of the findings

The comparison made is logical and data-oriented. The conclusions are well stated. It will be better if authors propose future work/an approach to overcome the third challenge or enhance AES systems.

Reviewer 3 ·

Basic reporting

The article is generally written in a clear, coherent and professional English. However, there are some segments that need to be corrected because they are not well understood or they are incorrectly written. It is necessary to improve the writing at lines:
- 41 to 43: they lack the sense of the rater as a person .... (improve the writing)
- 54 to 55: The extended-response, such as essays, … (it is not clear, improve the writing)
- 60 to 61: It occurs that the score of an essay scored … (improve the writing)
- 80: The most common models found for the AES systems are Natural Language Processing … (NLP is not a model, use another term to refer to NLP)
- 102-103: To end, a score is determined by estimating coefficients (β weights) from the training stage… (how the coefficients are calculated?)
- 234: It consists of 12.976 150-to-550 word-essays… (it is not clear, improve the writing)
- 298 to 299: 1->2, 3->4, 5->6 and 7->8, where 1->2 denotes prompt one as source domain and prompt. (I did not understand this notation, try to explain better)
- 336: Dasgupta and his colleagues employed in their experiments is that the ASAP… (improve the writing)
- 350 to 351: Recently, computer technologies, especially NLP and AI … (use a more appropriate term for NLP and AI)
- 421 to 423: … examined whether a computer could be deceived and assign a lower … (it is not clear, improve the writing)
- 425 to 426: one of the most recent technologies, which is Neural Networks… (use a more appropriate term for Neural Networks and improve the writing)

Figures:
- Figure 2: I think the figure is redundant. Maybe the information the figure contains could be explained in a simple list inside the paper content.
- Figure 3: The elements of the figure should be better explained.
- Figure 4: Both figures are equal and they do not show something special. You should use other example figures for a better understanding.

Table:
- Table 1: Spelling error of word “scorers” at the beginning of the table: Correlation with human scorers

Experimental design

no comment

Validity of the findings

no comment

Additional comments

The article is well written and structured. The Introduction and background are correct because they reveal the problems that need to be addressed and the objectives of the paper are clearly explained. Also, the methodology of this work is well developed. The conclusions are well established and they discuss the relevance of this research for the task of automate language essay scoring.

Some recommendations for the improvement of this paper are detailed in the basic reporting section.

---

## Round 0.2 · accepted · Accept

Your article is now Accepted. Congratulations

Reviewer 1 ·

Basic reporting

The authors addressed the issues raised, and the manuscript can now be accepted for publication.

Experimental design

No comment

Validity of the findings

No comment

Additional comments

The authors addressed the issues raised.

Reviewer 2 ·

Basic reporting

-

Experimental design

-

Validity of the findings

-

Additional comments

-